# Low excitatory innervation balances high intrinsic excitability of immature dentate neurons

Cristina V. Dieni[1,2], Roberto Panichi[2], James B. Aimone[3], Chay T. Kuo[4], Jacques I. Wadiche[1] & Linda Overstreet-Wadiche[1]

Persistent neurogenesis in the dentate gyrus produces immature neurons with high intrinsic excitability and low levels of inhibition that are predicted to be more broadly responsive to afferent activity than mature neurons. Mounting evidence suggests that these immature neurons are necessary for generating distinct neural representations of similar contexts, but it is unclear how broadly responsive neurons help distinguish between similar patterns of afferent activity. Here we show that stimulation of the entorhinal cortex in mouse brain slices paradoxically generates spiking of mature neurons in the absence of immature neuron spiking. Immature neurons with high intrinsic excitability fail to spike due to insufficient excitatory drive that results from low innervation rather than silent synapses or low release probability. Our results suggest that low synaptic connectivity prevents immature neurons from responding broadly to cortical activity, potentially enabling excitable immature neurons to contribute to sparse and orthogonal dentate representations.

[1] Department of Neurobiology and Evelyn McKnight Brain Institute, University of Alabama at Birmingham, Birmingham, Alabama 35294, USA. [2] Department of Experimental Medicine, Section of Physiology and Biochemistry, University of Perugia, Perugia 06126, Italy. [3] Data-driven and Neural Computing Department, Sandia National Laboratories, Albuquerque, New Mexico 87185, USA. [4] Department of Cell Biology and Neurobiology, Duke University School of Medicine, Durham, North Carolina 27710, USA. Correspondence and requests for materials should be addressed to J.I.W. (email: jwadiche@uab.edu) or to L.O.-W. (email: lwadiche@uab.edu)

The dentate gyrus (DG) has long been associated with the computational task of pattern separation, or the transformation of similar input patterns to output patterns that are less correlated. Sparse population coding is an essential component of pattern separation because the storage capacity for neural activity patterns is inversely related to the proportion of active principal neurons in a network[1–3]. Accordingly, sensory stimulation or spatial memory tasks activate only a small fraction of DG granule cells (GCs) ($<5\%$)[4–9]. Theories of DG pattern separation propose that strong inhibition selects small and distinct populations of active GCs in a manner that amplifies slight differences in inputs[2,10,11]. Remarkably, manipulating the small population of adult born GCs is sufficient to alter behaviours that require discrimination of similar contexts[12–16], leading to the idea that adult born neurons have an important role in pattern separation. However, it is unclear how GCs of various developmental stages contribute to DG network functions[15,16].

One way that newly generated GCs may contribute to DG function is if their distinct physiological properties confer unique contributions to coding processes. *Ex vivo* studies have shown that the morphological, intrinsic and synaptic properties of newly generated GCs undergo a protracted process of maturation during which immature GCs could transiently perform distinct network functions. Much attention has focused on a period when immature GCs are synaptically integrated within the circuit and also exhibit high intrinsic excitability that allow spiking in response to small current injections as well as distinctive integrative properties[17–19]. In this developmental stage that occurs ~4 weeks after cell birth, immature GCs also exhibit less synaptic inhibition than mature GCs, and afferent stimulation preferentially generates spiking in (that is, 'recruits') immature GCs over mature GCs[20–22]. Thus, in comparison to mature GCs, immature GCs appear to be highly excitable and broadly responsive, acting as good integrators of afferent activity[15,23,24].

However, it is not clear how the physiological properties of immature GCs identified *ex vivo* contribute to putative higher order functions *in vivo*. From a theoretical standpoint it is surprising that broadly responsive neurons contribute to the computational task of pattern separation since neurons that act as integrators reduce sparse population coding. In fact, inclusion of excitable immature GCs in a realistic network model degrades rather than improves pattern separation[23]. Furthermore, the contribution of immature GCs to network activity *in vivo* has been difficult to assess. Preferential recruitment of immature GCs *in vivo* was reported using cFos as a proxy for neural activation[6,7], but this conclusion has been disputed[8,25]. *In vivo* recordings from the DG of rodents has identified distinct functional populations with overall sparse patterns of activity in presumed principal cells[4,9]. But the interpretation of how immature GCs contribute to this activity has been conflicting, ranging from the possibility that immature GCs make up the entire population of active DG neurons[26] to more recent evidence that mature GCs are predominantly active during memory encoding[27].

At developmental stages when intrinsic excitability is high, immature GCs also exhibit features that suggest low glutamatergic synaptic connectivity, including small dendritic arbors, low spine densities and small evoked excitatory postsynaptic currents (EPSCs)[18,20,22,28–30]. Here we assess the role of synaptic connectivity in recruiting spiking in mature and immature GCs. Our results demonstrate that low excitatory connectivity from the entorhinal cortex (EC) prevents excitable immature GCs from spiking in response to afferent activity that is sufficient to generate spiking in mature GCs. Although immature GCs can spike with fewer active inputs than mature GCs (challenging the specific role of immature GCs in disambiguating input patterns), low innervation predicts that immature GCs sample a smaller component of EC afferent activity and thus exhibit lower correlations in synaptic inputs. Incorporating these results into a simple network model reveals that the combination of high excitability and low synaptic connectivity potentially provides an unexpected computational advantage wherein immature GCs enhance the range of EC activity levels that can be maintained with well-separated output representations.

## Results

**Immature GC spiking is limited by low excitatory drive.** In adult hippocampal slices, immature GCs that are approximately 4 weeks post mitosis are more likely than mature GCs to spike in response to stimulation in the molecular layer (ML), a paradigm in which synchronized perforant path excitatory drive is above spike threshold hence spiking is largely determined by synaptic inhibition and almost all cells spike when inhibition is blocked[20–22]. However, synchronous stimulation of a beam of perforant path axons in the ML may not provide excitatory drive representative of neuronal activity of EC projection neurons. Indeed, whole-cell (WC) recordings from mature GCs *in vivo* show a constant barrage of $\theta$-modulated asynchronous EPSCs arising from EC that is associated with infrequent spiking[31]. We thus sought to compare the relative spiking probably of mature and immature GCs in response to more diffuse afferent activity provided by direct stimulation of EC.

We used NestinCreER[T2] or NestinCreER[TM4] mice[22,32,33] crossed with Ai14 Cre reporter mice at 30–36 days post tamoxifen injection to identify immature GCs (Fig. 1a). Consistent with prior characterization of intrinsic and synaptic maturation of GCs in NestinCreER[T2] mice[22], at this interval tdTomato (tdT)-labelled (immature) GCs displayed repetitive spiking in response to current injections (Fig. 1b,c), a characteristic that develops after 3 weeks of neuronal maturation in retroviral labelled GCs[18]. Other intrinsic properties of immature GCs were consistent with ~4-week-old GCs identified by retroviral labelling[18,20,] whereas unlabelled GCs with fully mature membrane properties were classified as 'mature' (Fig. 1b,c). The persistence of some immature intrinsic properties in GCs recorded up to 36 days post tamoxifen injection is consistent with the slow rate of maturation in the ventral hippocampus from mice housed under standard conditions[30,34].

We examined spiking probability in response to simultaneous stimulation of the medial and lateral EC (MEC/LEC) across a range of stimulus intensities (Fig. 2a,b). This paradigm allows activation of the perforant path while avoiding direct stimulation of local interneurons[22], and also mimics GC integration of spatial and sensory information arising in the MEC and LEC, respectively. We previously showed that focal stimulation in the MEC alone generates EPSCs with paired-pulse depression whereas LEC stimulation evokes EPSCs with paired-pulse facilitation, and the amplitude of dual-pathway evoked EPSCs are nearly the sum of the individual EPSCs suggestive of independent pathways[22]. MEC/LEC stimulation generated spikes in 16% of mature GCs (5/32 cells) and blocking inhibition with gabazine increased the percentage of mature GCs that spiked to 50% (16/32 cells; Fig. 2b). Since MEC/LEC-evoked IPSCs are entirely blocked by NBQX[22], the gabazine-induced increase shows that feed forward inhibition contributes to GC sparse population activity[35]. Yet the fact that only 50% of GCs spiked in gabazine also shows that excitatory drive is a limiting factor for GC spiking in this stimulating paradigm, unlike stimulation in the ML, in which essentially all nearby GCs spike when inhibition is blocked[20,22]. Confirming this idea, stimulation in MEC/LEC

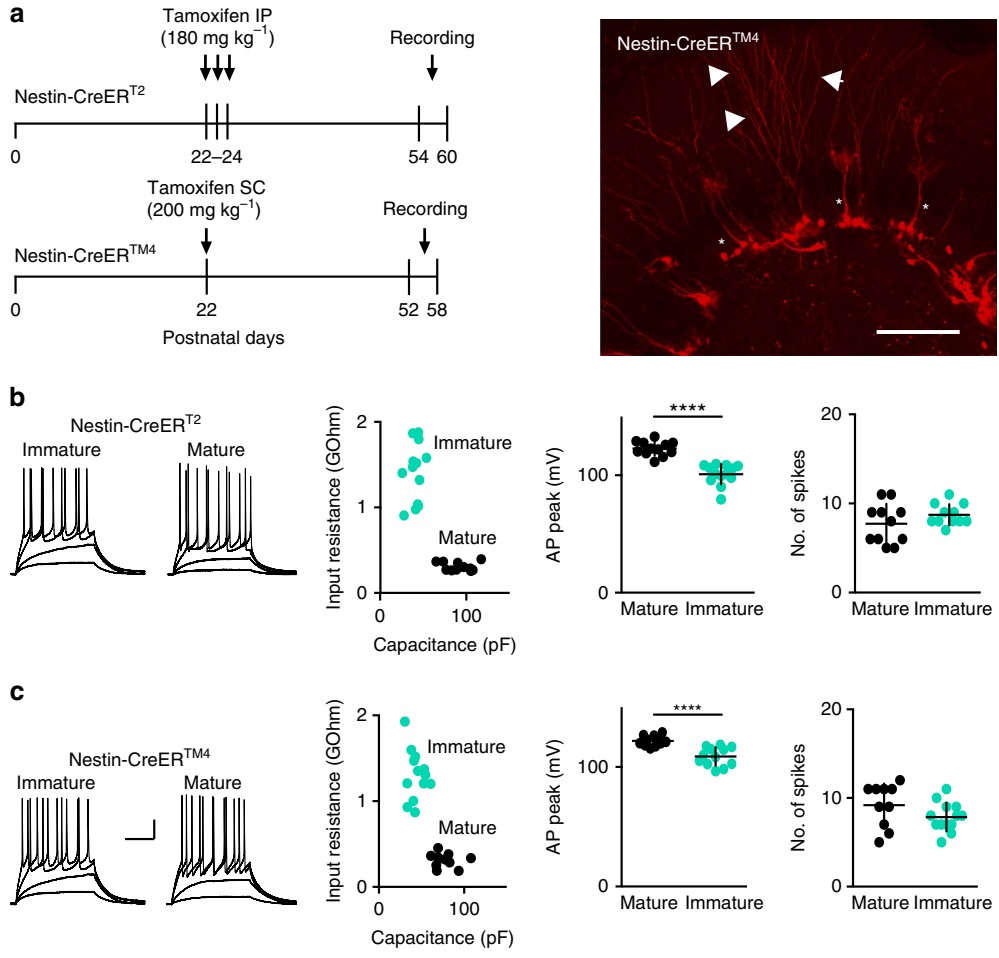

**Figure 1 | Identification of immature and mature GCs. (a)** Timeline of the experiments. (top) NestinCreER[T2] mice received three daily tamoxifen injections at P22-24 to induce tdT expression followed by recordings 30–36 days later. (bottom) Because of the lower efficacy of the NestinCreER[T2] line[58], some experiments were performed in NestinCreER[TM4] mice that received 1 day of tamoxifen injection at P22. (right) Confocal image showing tdT-labelled cells in a 50 μm section from a perfusion-fixed NestinCreER[TM4] mouse at 36 days PTI. Asterisks indicate putative type 1 cells with radial glial morphology. Immature GCs with dendrites projecting through the ML (arrowheads) were targeted for recordings. Scale bar, 100 μm. **(b)** (left) Examples of voltage responses to current injection in tdT-labelled immature GCs (10 pA current steps) and unlabelled mature GCs (20 pA steps) in NestinCreER[T2] mice. (right) There were significant differences in input resistance (1.40 ± 94 GΩ, $n = 13$ and 0.30 ± 14 GΩ, $n = 12$; $P < 0.0001$ unpaired $t$-test), membrane capacitance (40 ± 2 pF and 91 ± 4 pF, $P < 0.0001$) and the AP peak amplitude (101 ± 2 mV and 123 ± 2 mV, $P < 0.0001$), respectively[22], but no difference in the maximal number of spikes. **(c)** (left) Examples of voltage responses to current injection in tdT-labelled immature GCs (10 pA current steps) and unlabelled mature GCs (20 pA steps) in NestinCreER[TM4] mice. Cells had similar intrinsic properties as in **b**, with differences between immature and mature GCs in input resistance (0.31 ± 24 GΩ, $n = 11$ and 1.30 ± 80 GΩ, $n = 13$; $P < 0.0001$), the membrane capacitance (44 ± 3 pF and 77 ± 4 pF, $P < 0.0001$) and the AP peak amplitude (108 ± 2 and 122 ± 1 mV, $P < 0.0001$), but no difference in the number of spikes. Scale bars, 20 mV and 200 ms. PTI, post tamoxifen injection.

evoked smaller EPSCs and excitatory postsynaptic potentials (EPSPs) compared with ML stimulation, presumably due to the spread of fibres from the distal location of the stimulating electrodes and cut fibres in the slice (Supplementary Fig. 1).

Since many initial attempts to evoke synaptic responses in immature GCs by MEC/LEC stimulation were unsuccessful (not shown), we first identified synaptic input to a mature GC and then, without moving the stimulating electrodes, recorded from a neighbouring immature GC. Using this sequential analysis, we found that EPSCs evoked by the same MEC/LEC stimuli were dramatically smaller in immature GCs (Fig. 2c,d), and some immature GCs failed to respond altogether ($n = 4/18$; Supplementary Fig. 2a). Interestingly, EPSPs in immature GCs were likewise smaller than EPSPs in mature GCs (Fig. 2e), even when cell pairs with no input to the immature GC were excluded from analysis (Supplementary Fig. 2b). Addition of gabazine enhanced EPSPs in immature GCs (amplitude increased from

6.7 ± 1.3 mV to 8.1 ± 1.4 mV, $n = 6$, $P = 0.03$, Wilcoxon test), but still failed to elicit spikes despite the ability of immature GCs to spike with current injections (Fig. 2f). Thus, enhanced intrinsic excitability of immature GCs does not fully compensate for reduced excitatory drive in this paradigm[18]. Importantly, sequential recordings from two mature GCs using the same paradigm resulted in identical EPSCs/EPSPs in the second mature GC, confirming that neighbouring mature GCs were sampling synaptic inputs from the same population of active fibres and that small EPSCs in immature GCs did not result from optimizing the stimulation for the first mature GC or other experimental bias ($n = 9$ pairs of mature GCs; Supplementary Fig. 3). Again, 3/18 (16.6%) of these mature GCs displayed spiking, confirming the spiking probability of mature GCs in this paradigm.

The small EPSCs in immature GCs and failure to spike under conditions where mature GCs could be recruited to spike suggests that synaptic connectivity plays a crucial role in selecting active

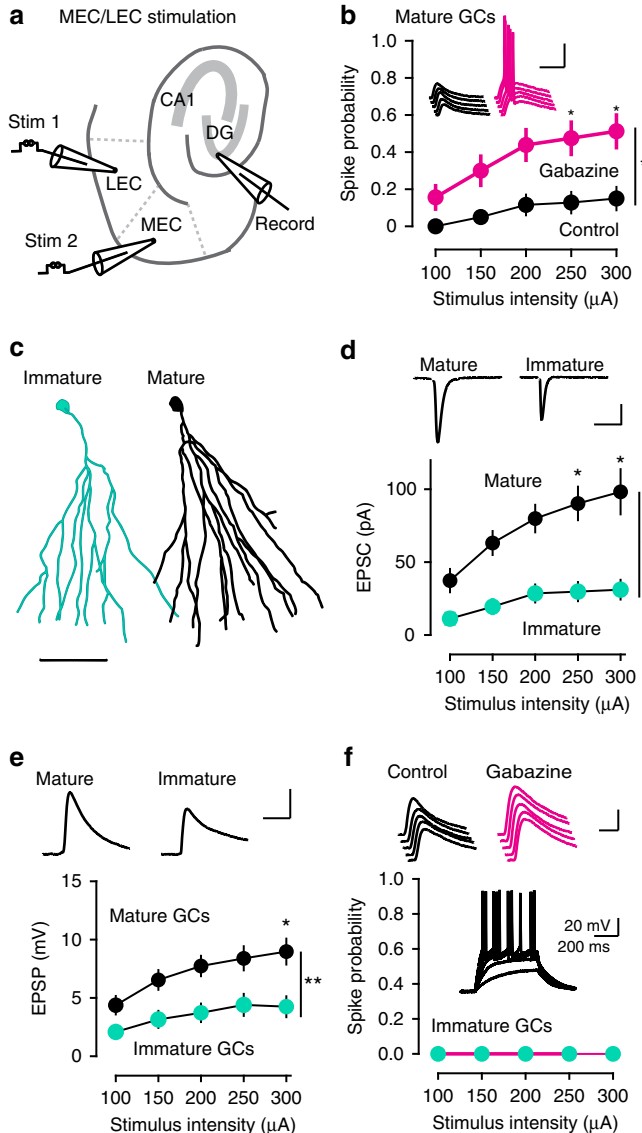

**Figure 2 | Preferential spiking of mature GCs in response to EC stimulation.** (**a**) Diagram of dual stimulation in the medial and lateral EC (MEC/LEC) in horizontal slices. (**b**) The fraction of spiking mature GCs was increased by gabazine (3 μM, magenta, $n = 32$). Repeated-measures two-way ANOVA: factor Gbz, $F_{(1,62)} = 12.35$, $P = 0.008$; factor stimulus, $F_{(4,248)} = 17.04$, $P = 0.0001$; Interaction, $F_{(4,248)} = 2.84$, $P = 0.025$, Tukey's post-test *$P < 0.05$; **$P < 0.01$. Inset, example EPSPs in control (black) and gabazine (magenta). Scale, 40 mV, 40 ms. All symbols are mean ± s.e.m. (**c**) Reconstructions of a typical immature (teal) and mature (black) GC, showing the smaller dendritic arbor of immature GCs[22]. Scale, 50 μm. (**d**) Sequential recordings from mature GCs (black) and immature GCs (teal) show smaller EPSCs in immature GCs ($n = 18$ pairs of mature and immature GCs). Scale, 20 pA, 50 ms. Repeated-measures two-way ANOVA: factor cell age, $F_{(1,34)} = 9.6$, $P = 0.004$; factor stimulus, $F_{(4,136)} = 12.8$, $P < 0.0001$; interaction, $F_{(4,136)} = 3.6$, $P = 0.008$; and Tukey's post-test *$P < 0.05$;**$P < 0.01$. (**e**) EPSPs were also smaller in the same mature and immature GC recordings ($n = 18$). Scale, 5 mV, 50 ms. Repeated-measures two-way ANOVA: factor cell age, $F_{(1,34)} = 10.18$, $P = 0.003$; factor stimulus, $F_{(4,136)} = 30.6$, $P < 0.0001$; interaction, $F_{(4,136)} = 3.3$, $P = 0.012$; Tukey's post-test *$P < 0.05$;**$P < 0.01$. (**f**) The spiking probability of immature GCs remained 0 at all MEC/LEC stimulus intensities, even when inhibition was blocked (magenta). Insets, EPSPs were enhanced by gabazine and all immature GCs could spike with current injections. Scale bar: 5 mV, 20 ms (top); and 20 mV, 200 ms (bottom). ANOVA, analysis of variance.

GCs. But to rule out the possibility that our WC recordings altered spiking behaviour by disrupting the intracellular milieu, we also examined spiking using noninvasive cell-attached (CA) recordings. Since we could not assess excitatory drive (synaptic responses) using CA recordings, we first used WC recording from a mature GC to confirm effective MEC/LEC stimulation. Then, without moving the stimulating electrodes, we assessed spiking using sequential CA recordings from multiple GCs within the field of view. In nine experiments where we evoked relatively large EPSCs monitored by WC recordings from mature GCs, we made a total of 57 CA recordings from nearby immature and mature GCs (Fig. 3a,b; note that there are many more mature GCs than immature GCs in each field of view). Similar to the WC recordings, 22% of mature GCs (9/41) exhibited spikes in CA mode, whereas none of the immature GCs displayed spikes (0/16; $\chi^2 = 4.1$, $P = 0.041$). The WC and CA results were not different, so we pooled all experiments to illustrate that immature GCs were less likely than mature GCs to spike in response to MEC/LEC stimulation (0/34 and 17/74, respectively, $P = 0.006$; Fig. 3b). Since immature GCs could spike in response to current injection, the failure to spike in response to EC stimulation resulted from insufficient excitatory depolarization. Indeed, comparing EPSPs and spiking probability illustrated that EPSPs in immature GCs generated by MEC/LEC stimulation were too small to achieve threshold (Supplementary Fig. 1c). Thus low excitatory drive can prevent spiking of immature GCs in response to MEC/LEC stimulation. The lack of spiking in this stimulating paradigm, however, does not mean that immature GCs fail to spike to any stimulus. In fact, preferential afferent-induced spiking of immature GCs indicates that they spike efficiently when they receive sufficient excitatory drive[18,20,22]. Rather, these results suggest that differential synaptic connectivity contributes to the spiking probability of mature and immature GCs.

**Reduced excitatory drive monitored by AMPA and NMDA EPSCs.** A potential caveat to the idea that immature GCs have less excitatory innervation than mature GCs is that newly generated GCs have silent synapses and a high ratio of *N*-methyl-D-aspartic acid receptors (NMDARs) to α-amino-3-hydroxy-5-methyl-4-isoxazolepropionic acid receptors (AMPARs) that may underestimate synaptic connectivity measured exclusively by AMPAR EPSCs[36,37]. We therefore assayed perforant path excitatory drive mediated by both AMPAR and NMDARs using simultaneous recordings of immature and neighbouring mature GCs during focal stimulation in the ML. In the presence of gabazine, we recorded AMPAR EPSCs at −70 mV, using the depressing and facilitating paired-pulse ratio (PPR) to confirm medial perforant path (MPP) or lateral perforant path (LPP) stimulation, respectively[22]. Consistent with Fig. 2, AMPAR EPSCs in immature GCs were smaller than in neighbouring mature GCs, for both MPP and LPP stimulation (Fig. 4a). Since the number of fibres activated by the stimulating electrode was the same for each mature/immature GC pair, the smaller EPSCs in immature GCs likely reflect fewer active synapses. We also blocked AMPARs with NBQX and found that NMDAR EPSCs recorded at −40 mV were likewise smaller in immature GCs during simultaneous recordings (Fig. 4b). Thus, low excitatory drive of immature GCs is apparent with both NMDAR EPSCs as well as AMPAR EPSCs. To assess potential silent synapses, we quantified the NMDAR/AMPAR ratio by comparing AMPA EPSCs at −70 mV and NMDAR EPSCs at +40 mV in the same cells during simultaneous mature and immature GC recordings. For MPP stimulation the ratio was significantly higher for immature GCs, consistent with a higher proportion of NMDAR to AMPARs on developing GC dendrites[36,37] (Fig. 4c; $n = 8$,

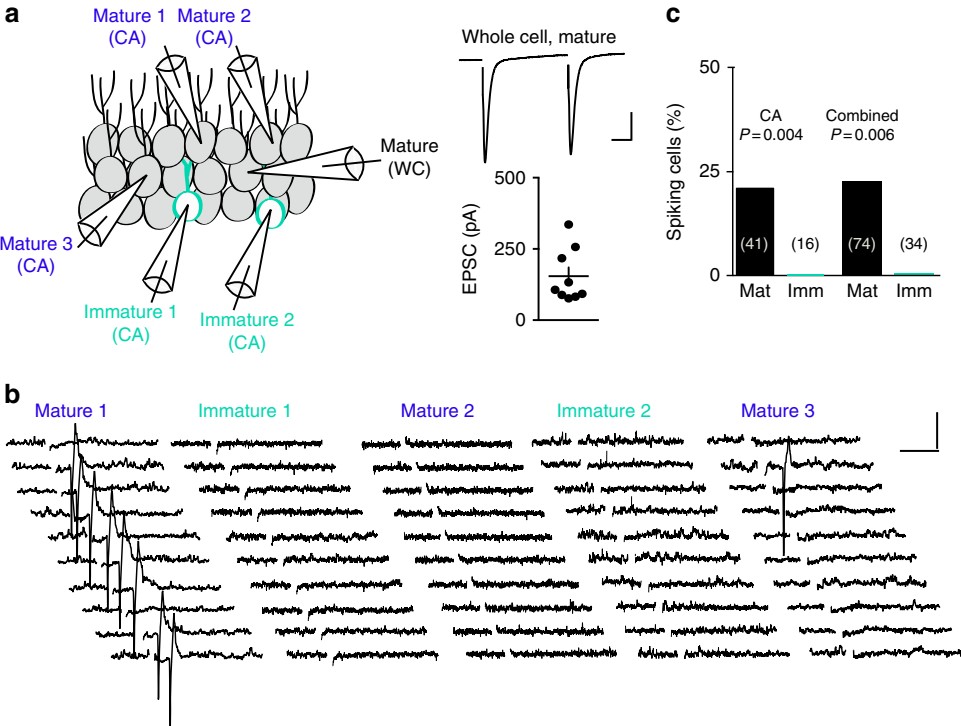

**Figure 3 | Lack of immature GC spiking using CA recordings.** (**a**, left) Cartoon depicting sequential CA recordings from multiple mature and immature GCs after confirming EPSCs in a WC recording from a mature GC (right). Scale bar, 100 pA, 50 ms. CA recordings were initiated after identifying relatively robust EPSCs in a mature GC ($n = 9$). (**b**) Examples of spiking in some mature GCs but not in immature GCs. The amplitude and variance of current was not different across cell age and there was no correlation between the noise and spike amplitude (not shown). Scale bar, 20 pA, 20 ms. (**c**) The fraction of spiking mature GCs was significantly higher than the fraction of spiking immature GCs, both for the CA recordings (left, 22% versus 0%; $\chi^2 = 4.1$) and the combined CA and WC data set (right, 23% versus 0%; $\chi^2 = 7.29$). The total number of recorded cells is indicated in parenthesis.

paired $t$-test; $P = 0.004$), whereas LPP stimulation generated a similar ratio ($n = 8$, paired $t$-test $P = 0.7$). These results suggest that MPP synapses with immature GCs have more silent synapses than mature GCs, but the potential confound of poor voltage control at distal synapses make it difficult to interpret the LPP results. Regardless, the small amplitude of pharmacologically isolated NMDAR EPSCs (Fig. 4b) suggest that silent synapses cannot account for low AMPAR-mediated excitatory drive.

**Release probability at mature and immature synapses.** Low excitatory drive to immature GCs could result either from fewer perforant path synapses (reduced innervation) or from low release probability ($P_r$) across a similar number of synapses. To differentiate these possibilities, we compared $P_r$ of perforant path synapses using the blocking rate of the NMDAR-EPSC by the irreversible open-channel blocker MK801 (ref. 38). After establishing a baseline of NMDAR EPSCs recorded at $-40$ mV (stimulating either the MPP or LPP in NBQX and gabazine), we applied MK801 (40 µM) for 5 min and then resumed stimulation to compare the rate of EPSC block at immature and mature synapses (Fig. 5). Repeated synaptic stimulation in the presence of MK801 provides a relative measure of $P_r$ since synapses with high $P_r$ are blocked faster than synapses with low $P_r$. The progressive block rate of NMDAR EPSCs was best described by two exponentials that we used to calculate a weighted decay time constant ($\tau_w$). For stimulation in the MPP, $\tau_w$ of NMDAR EPSCs in immature GCs was $15.8 \pm 2.1$ ms compared with $24.0 \pm 1.9$ ms in mature GCs ($n = 8$ each; unpaired $t$-test $P = 0.028$). The increased $\tau_w$ resulted from an increase in $\tau_{fast}$ with no change in $\tau_{slow}$ (Fig. 5a, inset; $\tau_{slow}$: $32.5 \pm 3.7$ compared with $37.2 \pm 6.1$, $n = 8$, $P = 0.32$ unpaired $t$-test), similar to what has been observed at

immature synapses in the developing hippocampus[39,40]. The blocking rate of immature and mature NMDAR EPSCs was not different in response to LPP stimulation (Fig. 5b; $\tau_w$: $19.9 \pm 1.8$ and $27.5 \pm 9.1$ ms, $n = 8$ each, $P = 0.44$; $\tau_{fast}$: $9.4 \pm 2.6$ versus $9.6 \pm 2.8$, $P = 0.98$; $\tau_{slow}$ : $35.4 \pm 5.2$ versus $42.7 \pm 5.7$, $P = 0.34$ unpaired $t$-tests). These results suggest that the release probability is higher rather than lower at immature MPP synapses. One potential caveat is that the MK801 blocking rate could be affected by different NMDAR subunit composition, since developing GCs have enriched expression of synaptic NMDAR2B receptors[36,41]. However, the 2B-specific antagonist R0-256981 (1 µM) blocked MPP-evoked EPSCs in mature and immature GCs by a similar degree (27% in mature and 32% in immature GCs; $n = 3$, $P = 0.7$ Wilcoxon–Mann–Whitney test) and the MK801-induced acceleration of the EPSC decay $\tau$, a measure of receptor open probability, was similar in mature and immature GCs (reduced by $31 \pm 6\%$ in mature and $20 \pm 4\%$ in immature GCs; $n = 8$, $P = 0.3$ paired $t$-test). Thus, immature GCs in our experiments have attained a largely mature complement of NMDARs.

The PPR also provides a relative measure of $P_r$ that can be assayed by AMPARs, and we found no differences in the PPR of AMPAR EPSCs in immature and mature GCs. The PPR of MPP-evoked EPSCs in mature GCs was $0.89 \pm 0.03$ compared with $0.84 \pm 0.03$ in immature GCs ($n = 10$ each, $P = 0.3$, paired $t$-test), and the PPR of LPP-evoked EPSCs in mature GCs was $1.16 \pm 0.05$ compared with $1.22 \pm 0.08$ in immature GCs ($n = 10$, $P = 0.4$, paired $t$-test). Thus, the PPRs and MK801 blocking rates show that limited excitatory drive to immature GCs does not result from low $P_r$ at a similar number of synapses.

**Low overlap in perforant path synaptic inputs.** As previously reported[18], we found that the frequency of mEPSCs in immature

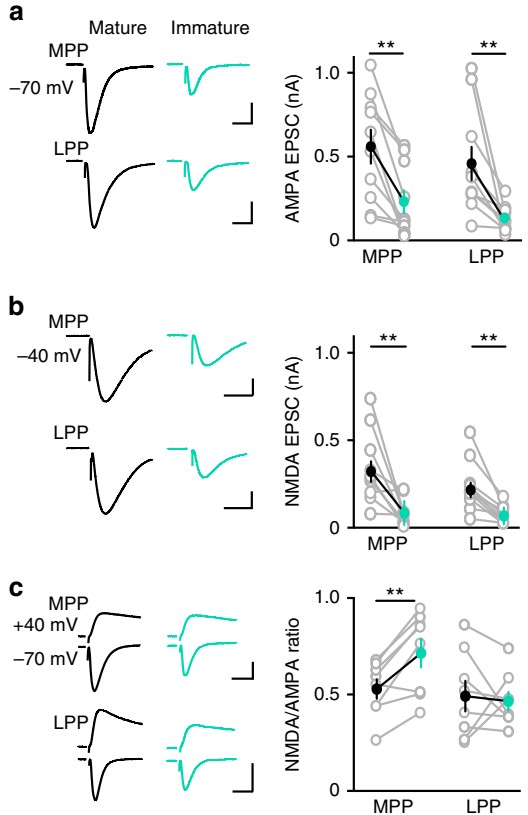

**Figure 4 | Reduced excitatory drive monitored by AMPA and NMDA EPSCs.** (**a**, left) Simultaneous recordings of AMPA EPSCs in mature (black) and immature GCs (teal) evoked by stimulation of the MPP (top) or LPP (bottom). (right) Values from simultaneous recordings are connected by lines. EPSCs in the MPP were 556 ± 101 pA and 228 ± 69 pA, n = 10. EPSCs in the LPP were 455 ± 100 and 130 ± 25 pA, n = 10. Paired t-test **P = 0.001. Solid symbols are mean ± s.e.m. Scale bar, 200 pA, 50 ms. (**b**) Simultaneous recordings of NMDA EPSCs in mature (black) and immature GCs (teal) evoked by stimulation of the MPP (top) or LPP (bottom). EPSCs in the MPP were 343 ± 64 pA and 85 ± 24 pA, n = 10. EPSCs in the LPP were 239 ± 46 pA and 77 ± 16 pA, n = 10, Paired t-test **P = 0.009. Scale bar, 100 pA (top) or 50 pA (bottom), 100 ms. (**c**, left) NMDA EPSCs (top, + 40 mV, NBQX) and AMPA EPSCs (bottom, − 70 mV) in simultaneous recordings of mature and immature GCs. (right) The NMDA/AMPA ratio of the MPP EPSC was larger in immature GCs (0.71 ± 0.07 compared with 0.53 ± 0.05, n = 8, Paired t-test **P = 0.004). Scale bar, 100 pA (top) or 50 pA (bottom), 50 ms.

GCs was lower than in mature GCs (0.52 ± 0.15 Hz versus 1.65 ± 0.25 Hz, n = 5, P = 0.008) with no difference in amplitude (6.25 ± 0.58 pA versus 6.88 ± 0.84 pA, P = 0.87). Together these results indicate that immature GCs receive less innervation from the perforant path compared with mature GCs, further suggesting that immature GCs retain high synaptic specificity since they sample only a fraction of the afferent axons arising from the EC population. To further assess this idea, we tested the ability to evoke EPSCs in immature and mature GCs using a modified paradigm previously used to define fine-scale specificity of cortical synaptic connectivity[42]. We compared the probability of evoking EPSCs in simultaneously recorded pairs of GCs, using low-intensity stimulation of the MPP or LPP to activate small numbers of perforant path axons. We identified a stimulation location where focal stimulation (2 μA) was just sufficient to reliably evoke an EPSC in one GC, and then we quantified the percentage of trials where an EPSC was generated in the second

GC as a function of stimulus intensity (Fig. 6a), where the % simultaneous success is defined as (number of trials with an EPSC in both cells/total number of trials) × 100. For recordings from two mature GCs or a mature and an immature GC, somata were located within 80–120 μm of each other. For pairs of mature GCs, increasing the stimulus intensity steeply increased the percentage of simultaneous successes (Fig. 6b,c, black symbols). This suggests that a largely overlapping population of afferent fibres innervate distinct mature GCs, that is, mature GCs have low synaptic specificity because there is high probability of activating fibres that synapse onto both cells as the stimulus intensity is increased. However, the percentage of simultaneous successes was significantly lower for pairs of a mature GC with an immature GC (Fig. 6b, teal symbols). At the lowest stimulus intensity, EPSCs were always observed in mature GCs with failures in immature GCs, consistent with lower innervation of immature GCs. There was no difference in the latency of EPSCs and the amplitude of EPSCs increased linearly with stimulation intensity, suggesting that increased stimulation recruited additional inputs to both cells rather than a separate population of inputs to the second cell (Supplementary Figs 4 and 5). The similar amplitude of successes at the lowest stimulus intensity also suggests that postsynaptic sensitivity (that is, receptor number) at immature synapses does not account for reduced excitatory drive (Supplementary Fig. 4), in accordance with the similarity of mEPSC amplitudes noted above. Furthermore, pairs of two immature GCs with somata within 80 μm of each other likewise displayed lower percentage of simultaneous successes compared with mature GC pairs within the same slices for both MPP and LPP stimulation, confirming that the overlap in synaptic input to immature GCs is lower than the overlap in input to mature GCs (Fig. 6c). Finally, we found that simultaneous recordings of medial and lateral perforant path-evoked EPSCs between immature GCs at 39–52 days post tamoxifen treatment and neighbouring mature GCs displayed similar % successes as two mature GCs (Supplementary Fig. 6). As described previously, many immature GCs at this later developmental stage exhibit excitatory synaptic currents, intrinsic excitability and spiking behaviour that approaches mature values[22]. Thus as synaptic innervation progresses across the first 2 months of new GC maturation[18], overlap in synaptic input also increases.

Reduced overlap in synaptic inputs could occur either if immature GCs receive fewer synaptic contacts per EC fibre, or if immature GCs receive innervation from a smaller number of fibres. It is difficult to discriminate these possibilities because the large variance of quantal parameters in GCs obscures quantal analysis of evoked EPSCs[43]. However, we favour the latter option because the former requires that unitary EPSCs in mature GCs are generated at synapses comprised of many release sites. The small amplitude of K+-evoked unitary EPSCs from perforant path is consistent with only a few release sites[35], supporting the idea that excitatory projection cells typically innervate each other at a small number of sites[44,45]. Furthermore the amplitude of sEPSCs in mature GCs was similar to the amplitude of mEPSCs (in the same cells, P = 0.1, paired t-test, n = 10), and the small amplitude of low-intensity-evoked EPSCs (Supplementary Fig. 5) further support small numbers of release sites per fibre. Thus, we predict that that immature GCs sample the activity of fewer perforant path fibres (and EC projection neurons) than mature GCs.

**Simulation of distinct connectivity in network functions.** It is generally thought that DG contributes to hippocampal memory encoding by orthogonalizing cortical activity patterns using very sparse population coding. Paradoxically, immature GCs with high

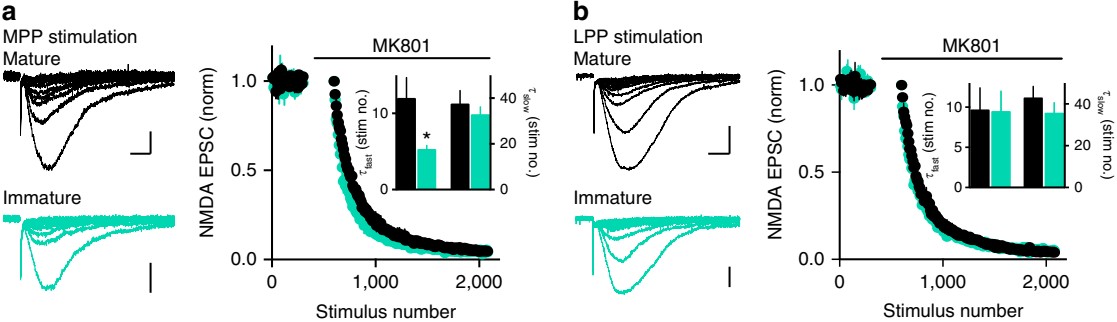

**Figure 5 | Release probability at mature and immature synapses.** (**a**, left) Examples of progressive block of MPP-evoked NMDA EPSCs in mature (black) and immature GCs (teal). Every 10th EPSC is shown. Scale bars, 50 pA, 20 ms. (right) The normalized amplitude of NMDA EPSCs is plotted against stimulus number and fit by two exponentials. The fast component ($\tau_{fast}$) was faster in immature GCs ($\tau_{fast}$: 5.2 ± 0.6, $n = 8$, compared with 11.9 ± 2.8, $n = 8$, $P = 0.037$, unpaired $t$-test), suggestive of higher release probability. Grouped data include both simultaneous recordings from mature and immature GCs ($n = 5$ cell pairs) and individual GC recordings ($n = 3$ cells, total of $n = 8$). (**b**, left) Examples of MK801 block of LPP-evoked NMDA EPSCs. Scale bar: 100 pA (top), 20 ms; or 50 pA (bottom), 20 ms. The blocking rate was the same in mature and immature GCs ($n = 8$ each, $P = 0.98$).

intrinsic excitability and low inhibition (that are preferentially recruited by afferent activity) are expected to reduce population sparseness and degrade pattern separation[23]. Our results suggest that low excitatory innervation could limit the recruitment of immature GCs into active neural ensembles, thereby counteracting neurogenesis-induced degradation of orthogonalization. To test this idea, we fit the experimental data shown in Fig. 6 to a simple statistical model designed to assess the overlap in GC output patterns across different levels of EC input (Fig. 7; see Methods). To isolate the contribution of differing levels of excitatory connectivity, the model did not include inhibition (which is known to differ between mature and immature GCs[20–22]). We assumed that the same aged GCs have equivalent number of synapses that sample input from the same total set of perforant path fibres (with $P_r = 1$). We constrained the ratio of excitatory connectivity (modelled as the number of synapses) to GCs according to the average ratio of EPSC amplitudes that we measured in simultaneous recordings of immature and mature GCs over a large range of stimulus locations and intensities (0.35; Fig. 7a). The EPSC amplitude depends on the number of fibres activated by the stimulating electrode, the release probability of each fibre, the quantal size and the number of active fibres that innervate each cell. In these experiments, the number of fibres activated by the stimulating electrode, the presynaptic release probability and mEPSC amplitude are the same for mature and immature GCs, thus the relative size of EPSCs reflects the likelihood that activate fibres innervate each cell. Assuming random connectivity, binomial statistics can be used to estimate the density and overlap of synaptic connectivity for mature and immature neurons. Using a fitting approach (see Methods), we observed strong fits with a $p_{minimal}$ of 0.39% and $N_\infty$ of 1,296 fibres for MPP stimulation. For mature GCs, we estimated an average of 219 MPP inputs, of which 37 were shared between pairs of neurons, and for immature GCs we estimated 77 inputs, of which five were shared (note that this analysis is meant to replicate our experimental paradigm rather than to recapitulate total synapse number). Most combinations of input values with good fits provided similar outcomes (the top 20% of random parameter fits are shown in Supplementary Table 1). We then generated networks of neurons obeying these statistics, and observed that the randomly connected neurons exhibited comparable overlapped synaptic inputs as observed experimentally (Fig. 7b). Next, we generated a simple network similar to that used previously to simulate DG function[21], whereby different GC neurons had connection densities representative of either all

mature, all immature or a mixture of both. These networks shared input connection statistics comparable to the observed slice results. To incorporate the higher intrinsic excitability of immature GCs, we dictated that each GC would fire if 20% of their synapses were active, allowing immature GCs to fire with lower numbers of active synaptic inputs[17,18].

Our results and others suggest that relative spiking of mature and immature GCs depended on the strength of the input, thus we did not vary the input correlations (keeping the inputs random) but rather examined the effects of different input levels on the output correlations. Using this approach, we assessed how differing levels of afferent stimulation (corresponding to different levels of EC activity) affects the overlap in GC output. Consistent with previous modelling[23], networks with immature neurons exhibited higher correlations (reduced orthogonality) than networks without neurogenesis when the input activity was on average below threshold for GCs to fire (dotted line in Fig. 7c). Recruitment of excitable immature GCs decreases sparseness and increases output overlap, apparently detrimental to the proposed role of immature GCs pattern separation[23,46]. In this input range, the response curve of the mature only network was steep; if EC activity was well below threshold, the mature-only DG could orthogonalize within a limited range of active EC inputs, but the network became ineffective as threshold was approached. Interestingly, networks of all immature neurons more gradually increased overlap as input activity approached threshold, and also displayed reduced overlap at higher levels of EC activity. Thus, excitable but poorly connected immature neurons are less sensitive to changes in input levels (green line, Fig. 7c), potentially suggesting that different mixtures of immature and mature neurons could regulate the range of tolerable EC activity levels.

To assess how neurogenesis affects the range of input levels than can be maintained with low overlap in outputs, we further tested how the network responded across a large spectrum of young neuron densities (0–100% immature neurons) with EC activity levels (0.1–0.22 of EC neurons active). This analysis requires that we define a tolerable range of overlap, which we set as the difference between the EC activity level that provided at least a normalized dot product (NDP) of 0.005 and the EC level that provided at least an NDP of 0.05 (that is, between 0.5 and 5% overlap). Although somewhat arbitrary, this low range of overlap is consistent with the generally accepted idea that the point of pattern separation in the DG is to provide near-orthogonal inputs to downstream CA3 (ref. 10).

Figure 7d illustrates the responses of four networks with different fractions of immature neurons with the tolerable range

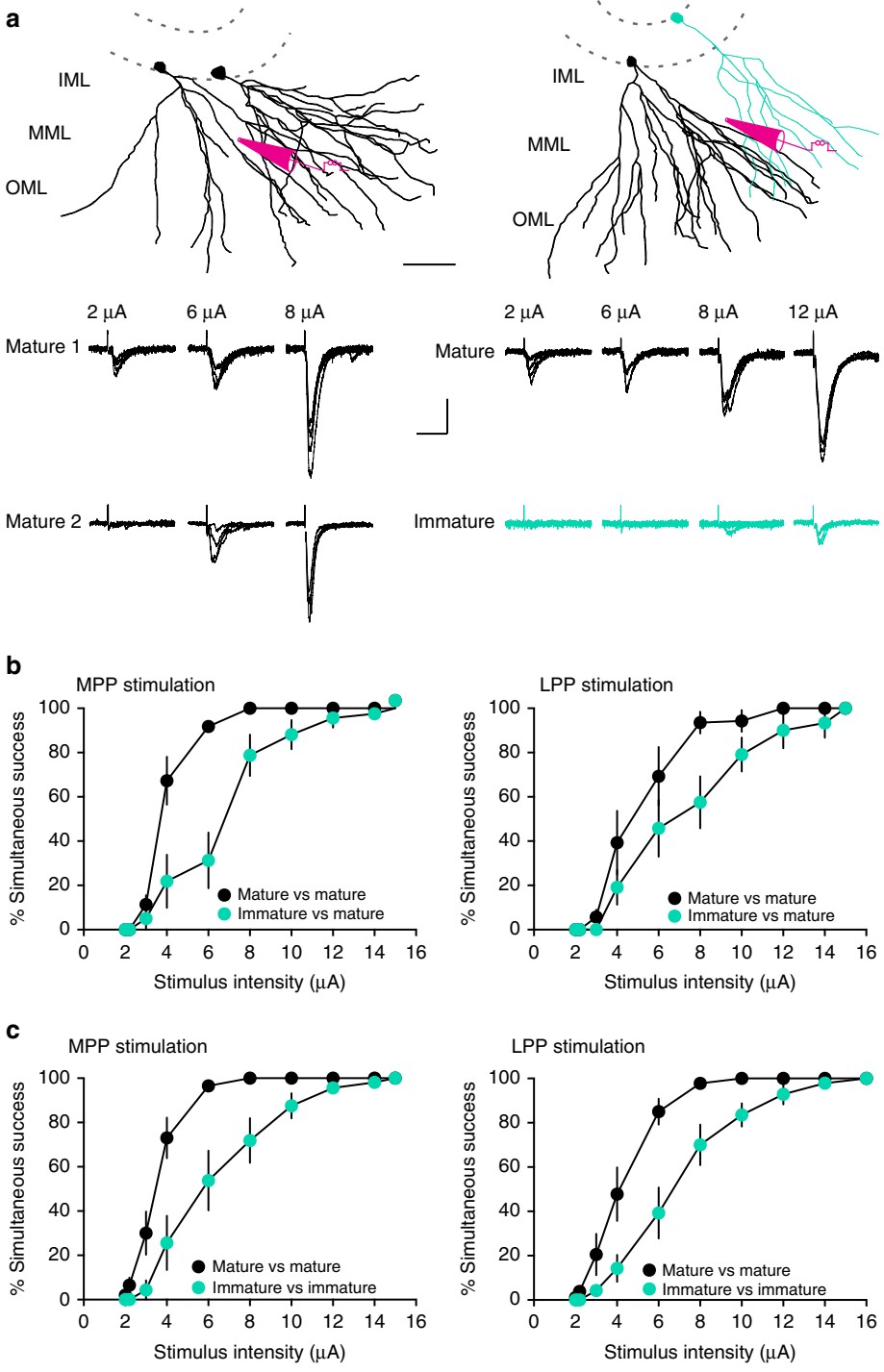

**Figure 6 | Less overlap of synaptic inputs to immature GCs.** (**a**, top) Reconstructions of two mature GCs (black, left) or a mature and an immature GC (teal, right) showing the approximate placement of the stimulating electrode. Scale bar, 50 μm. (bottom) The percentage of trials with simultaneous EPSCs in both cells (% simultaneous success) was measured at increasing stimulus intensities (20 trials at each intensity). Examples are from MPP stimulation. Scale bar, 20 pA, 40 ms. (**b**) The percentage simultaneous success versus stimulus intensity is shown for MPP stimulation (left) and LPP stimulation (right). The percentage simultaneous success for pairs of mature with immature teal symbols) was different from pairs of two mature GCs (black symbols) at multiple stimulation intensities. This suggests more active fibres were required to recruit overlapping synaptic inputs in pairs with an immature GC. Repeated measure two-way ANOVA, MPP, $n = 8$–11 cell pairs per group: factor cell age, $F_{(1,17)} = 14.6$, $P = 0.0013$; factor stimulus, $F_{(9,153)} = 173.2$, $P < 0.0001$; Interaction, $F_{(9,153)} = 10.6$, $P < 0.0001$. LPP, $n = 6$–7 cell pairs per group: factor cell age, $F_{(1,11)} = 6.3$ $P = 0.04$; factor stimulus, $F_{(9,99)} = 98.7$, $P < 0.0001$; Interaction, $F_{(9,99)} = 3.2$, $P = 0.031$. (**c**) The % simultaneous success for pairs of two immature GCs (teal) was different from pairs of two mature GCs (black symbols) at multiple stimulation intensities, indicated less overlap in synaptic inputs between immature GCs. Repeated measure two-way ANOVA, MPP, $n = 10$–8 cell pairs per group: factor cell age, $F_{(1,16)} = 16.1$, $P = 0.001$; factor stimulus, $F_{(9,144)} = 131.9$, $P < 0.0001$; and interaction, $F_{(9,144)} = 6.11$, $P < 0.0001$. LPP, $n = 9$–7 cell pairs per group: factor cell age, $F_{(1,14)} = 16.7$, $P = 0.001$; factor stimulus, $F_{(9,126)} = 138.2$, $P < 0.0001$; and interaction, $F_{(9,126)} = 4.8$, $P < 0.0001$. ANOVA, analysis of variance. Inner molecular layer (IML); middle molecular layer (MML); outer molecular layer (OML).

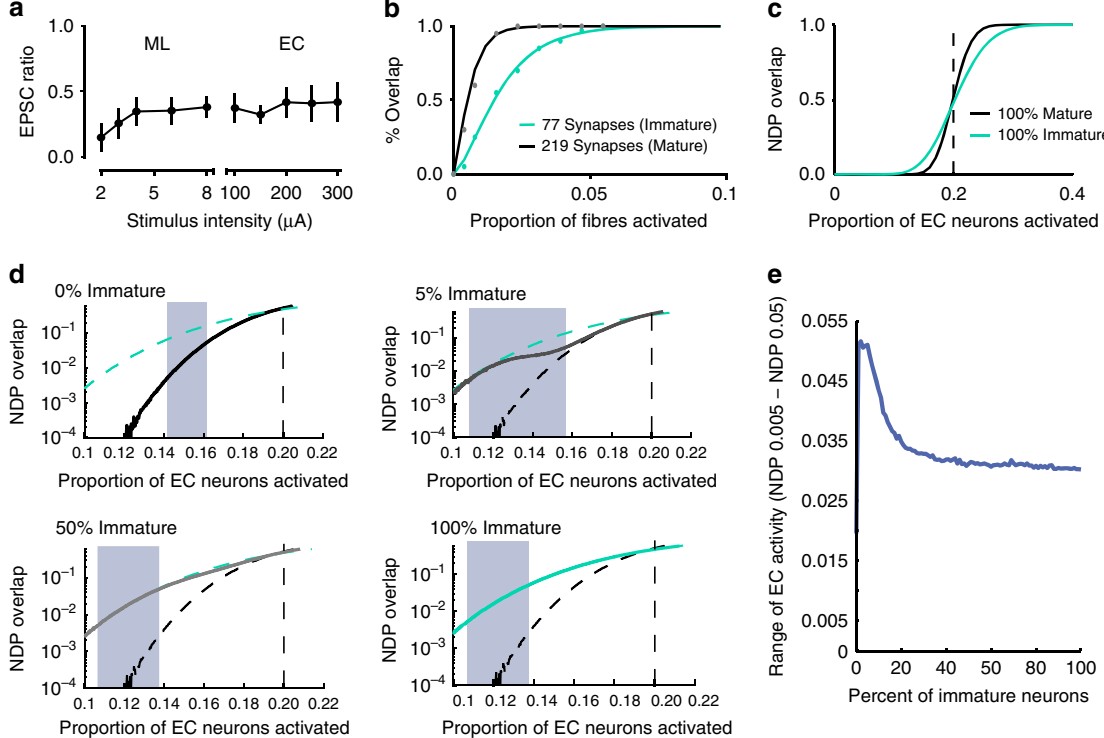

**Figure 7 | Immature neurons expand the input range for low-output overlap.** (**a**) The average ratio of EPSC amplitudes in immature and mature GCs was constant across a large range of stimulus intensities. EPSC ratios were calculated from simultaneous recordings of immature and mature GCs in response to ML stimulation ($n = 16$) and EC stimulation ($n = 18$). (**b**) Data from Fig. 6c (dots) with fits to equation (3) (lines). The total number of potential inputs ($\sim 1,300$ mPP) and average number of synapses on mature ($\sim 219$) and immature ($\sim 77$) GCs were found to achieve this fit. Results were robust to many fits, shown in Supplementary Table 1. (**c**) The average output correlations (NDP) of networks with mature and immature neurons are differentially affected by varying the level of EC activity (that is, the proportion of active EC neurons). (**d**) Expanded view of **c** describing the range of EC levels that can maintain low NDP overlap in networks with different percentage of immature GCs. In each panel, the network response is represented by the solid line and hypothetical limit conditions are represented by dashed lines. The percentage of immature GCs are indicated by different colours: black, 0% immature GCs; and teal, 100% immature GCs. The grey line is the response of the network with the indicated mixture of immature and mature GCs. Shaded areas correspond to the range of EC levels that maintain the network between 0.005 and 0.05 NDP overlap. (**e**) Plot of the input range (as in **d**) for networks containing from 0 to 100% of immature GCs. A network with a low percentage of immature neurons ($<5\%$) can tolerate the broadest range of active EC inputs.

of NDP highlighted by blue shading. Networks without immature neurons showed a small range of allowable input levels, with little difference between the level of EC inputs that was insufficient to drive DG activity and levels that induced high correlations. In contrast, networks of all immature neurons and those with equal mix of immature and old showed a larger range of allowable input levels given the more gradual recruitment of immature neurons into activated populations. Notably, the greatest range of allowable input levels was observed for networks that were mostly mature with a small fraction of immature neurons. In effect, these networks captured the best of both populations; the upper bound of permissible input level was increased by the lower excitability of the mature neurons and the gradual recruitment of immature neurons, whereas the lower bound of permissible inputs was reduced by excitable immature neurons that can be recruited by the small number of active inputs. This effect is seen more directly when the tolerable range of inputs for all neurogenesis levels from 0 to 100% are compared (Fig. 7e).

## Discussion

Here we assess the role of excitatory drive in afferent-induced spiking of immature and mature GCs. First, we show that the relative probability of perforant path-induced spiking depends on the stimulus paradigm. Although strong beam-like stimulation

paradigms with supra-threshold excitatory drive generate preferential spiking of immature GCs because of their reduced inhibition[20–22], we found that weaker (and potentially diffuse) stimuli preferentially recruited mature GCs. Our results suggest that low excitatory drive from the perforant path provides a previously underappreciated mechanism that prevents broad responsiveness of immature GCs. Second, we show that low excitatory drive to immature GCs results from less innervation rather than functional differences at immature synapses. Low innervation is consistent with the low frequency of mEPSCs and low spine density in retroviral labelled immature GCs[18,47] as well as the small dendritic trees of transgenic-labelled immature GCs[22]. Finally, we extend our experimental results to predict how poorly connected immature GCs could contribute to network functions. Using a simple statistical model, we show that excitable immature GCs with low innervation enhance pattern overlap at low input levels yet decrease pattern overlap at high input levels, potentially enhancing the range of input levels that can maintain well-separated output representations. Together these results suggest that low innervation counteracts high intrinsic excitability and contributes to distinct input–output transformations than expected for high excitability alone.

Our results suggest that the small dendritic structure of developing GCs has functional significance in limiting innervation. Since immature GCs are in a transient period of

cell growth, the magnitude of excitatory drive is correlated with morphological maturation and post-mitotic cell age[18,22,29,30,48]. Functional and morphological maturation of newly generated GCs is heterogeneous as well as progressive, and depends on diverse factors including animal age, housing condition, septal-temporal location and local network activity[19,30,34,48–50]. In young adult rodents, newly generated GCs exhibit relatively rapid dendritic and spine development during the first month after cell birth that continues over many subsequent weeks and is paralleled by the development of functional excitatory synapses[18,22,28]. Since developing GCs progress through immature stages when innervation is inversely correlated with intrinsic excitability[22], our conclusions are relevant to understanding immature GC function across various stages of maturation with the caveat that the timing of a particular stage varies according to specific conditions. Developing GCs appear to undergo the same sequence of maturation regardless of the age of the animal, thus we also expect that the factors contributing to immature GC spiking in young adult mice are relevant to understanding the function of immature GCs generated in older adult mice that are typically used to assess behavioural consequences of DG neurogenesis.

It is proposed that immature GCs are better integrators of afferent activity than mature GCs because of enhanced intrinsic excitability and reduced inhibition that enables preferential afferent-induced spiking[15,20,22,23]. Our results suggest an alternative view that low innervation counteracts broad responsiveness, an outcome that is not evident from experiments using afferent stimulation in which synaptic excitation is well above threshold[20,22]. Since cortical excitatory cells typically innervate each other at a small number of synaptic contacts[44,45], low innervation suggests that immature GCs sample the activity of a significantly smaller fraction of EC projection neurons than mature GCs. Low sampling of EC is consistent with the observed low overlap in synaptic inputs between immature and mature GCs, as well as between neighbouring immature GCs. Low sampling could enhance response selectivity by permitting immature GCs to integrate the activity from a more restricted population of EC projection neurons than mature GCs, even as high intrinsic excitability and weak inhibition promote integration of the activity arising from that population. Thus we speculate that immature GCs have higher input selectivity than mature GCs, based on the observation in the visual system that small dendritic arborization (that is, low sampling of synaptic inputs) correlates with relatively high selectivity within a class of neuron[51]. *In vivo* recordings from age-identified GCs will be required to assess this prediction.

It is important to reiterate that the lack of immature GC spiking in response to EC stimulation in our experiments does not mean that immature GCs are functionally silent. Immature GCs display preferential spiking under conditions of strong excitatory drive[20,22], indicating that they spike efficiently with sufficient excitation. Rather, our results support the idea that sparse activity in mature and immature cells rely on distinct mechanisms, with inhibition strongly contributing to sparse mature GC activity whereas low excitatory innervation contributes to sparse immature GC activity[22]. Extending the interpretation to *in vivo* functions is subject to the caveat that slice experiments compare spiking probabilities in response to synchronous activation of afferent fibres in a manner that is useful to assess relative synaptic connectivity and responsiveness across cell types/ages, but that may not fully predict how neurons respond to complex spatial-temporal patterns of afferent activity generated by sensory and spatial stimuli *in vivo*. Additional experiments will also be required to assess the role of hilar mossy cell innervation in GC spiking[52,53].

Comparing our experimental results with the simulation requires qualifications. The model implies that immature GCs mediate most if not all network activity at the very lowest EC input levels, ostensibly conflicting with the experimental data showing preferential recruitment of mature GCs in the EC stimulating paradigm. Importantly, the EC activity level of the experiments corresponds to an EC input level above the dotted line, since many mature GCs spike (in the absence of inhibition, Fig. 2b). In this range, the lower NDP overlap in the 100% immature network (Fig. 7c) implies a smaller fraction of active immature cells (compared with the 100% mature network) and is thus qualitatively consistent with the experimental data. At this EC activity level there is no effective orthogonalization since NDP is well outside the 'tolerable' range and it is insensitive to small percentages of immature GCs (Fig. 7d). This raises two major caveats of the model. First, the model assays output overlap as a function of input level rather than input correlations, thus it is not a conventional measure of pattern separation defined as the transformation of similar input patterns to output patterns that are less correlated. Second, the lack of inhibition in the model compresses the range of inputs that can be orthogonalized, since synaptic inhibition provides input normalization that allows networks to respond to a wide range of inputs without saturation[54]. Thus the model primarily serves to illustrate the main point that immature GCs with high excitability and low synaptic connectivity differentially affect NDP overlap across input levels, whereas high intrinsic excitability alone would be predicted to reduce population sparseness (and increase NDP overlap) across all input levels.

Our results can be incorporated into a broader view of the DG's function in hippocampal coding[10,55]. If the DG is relevant in driving CA3, it is necessary that its outputs have some minimal level of activity—perfect separation is meaningless if no information is communicated. Even a very low activity level necessitates some minimal level of neuronal overlap; however, too much overlap presumably leads to interference in CA3 memory formation. In our simple model, maintaining low overlap requires that the EC's activity level can only be tolerated within a small range that is more than doubled by the inclusion of immature neurons (Fig. 7d). The low percentage of immature GCs that is optimized for expanding this range could imply that small numbers of excitable but sparsely innervated immature neurons facilitate input–output transformations by promoting discrete network representations across variable levels of EC activity.

Finally, it is important to consider that 'what' young and mature neurons encode is likely as important as 'how' they encode it[15]. One implication of differential synaptic connectivity is that old and young GCs could represent different aspects of information incoming from EC. Because they are sampling more cortical space, mature GCs may encode and separate based on complex characteristics formed by many features of representation. Because of their limited sampling of EC, immature GCs may codify selective features of a representation with high fidelity due to their intrinsic excitability and low inhibition. In this manner, immature GCs could encode a singular aspect of a representation, potentially providing selectivity within fewer dimensions, which may help in contextualizing information incoming from EC based on combinations of concurrent spatial or temporal features[56,57]. Thus immature GCs in the network could increase memory resolution or acuity as well as contribute to associating a contextualizing event to CA3 during memory formation[23].

## Methods
We used male and female ~8-week-old tamoxifen-inducible nestin-based reporter mice. Nestin-CreER[T2] mice[32] were maintained on the C57Bl/6J background

(Jackson Labs, # 016261) and Nestin-CreER$^{TM4}$ (provided by Kuo et al.)[33] were maintained on the CD1 background. Both lines were crossed with Ai14 reporter mice (Jackson Labs, # 007914) to obtain offspring used in experiments. All animal procedures followed the *Guide for the Care and Use of Laboratory Animals*, the US Public Health Service, and were approved by the University of Alabama at Birmingham Institutional Animal Care and Use Committee. Mice were maintained in standard housing (2–5 per cage) in a 12:12 h light:dark cycle.

Nestin-CreER$^{T2}$ mice were injected with tamoxifen at 180 mg kg$^{-1}$ d$^{-1}$ for 3 days (IP) dissolved in 10%EtOH/90% sunflower oil[22,32]. Nestin-CreER$^{TM4}$ mice were injected with a single dose of tamoxifen at 8 mg per 40 g (SC) dissolved in 100% sunflower oil (20 mg ml$^{-1}$) (ref. 33). Tamoxifen treatment was initiated after weaning at P22. Mice were anaesthetized and perfused intracardially with ice-cold modified artificial cerebrospinal fluid (ACSF) containing the following (in mM): 110 choline chloride, 26 D-glucose, 2.5 MgCl$_2$, 2.5 KCl, 1.25 Na$_2$PO$_4$, 0.5 CaCl$_2$, 1.3 Na-ascorbate, 3 Na-pyruvate and 25 NaHCO3, bubbled with 95% O$_2$/5% CO$_2$. The brain was removed and 300-µm-thick horizontal slices were prepared using a vibratome (Leica VT1200, Leica Instruments). Slices were incubated at 37 °C for ~30 min in recording solution containing the following (in mM): 125 NaCl, 2.5 KCl, 1.25 Na$_2$PO$_4$, 2 CaCl$_2$, 1 MgCl$_2$, 25 NaHCO$_3$ and 25 D-glucose bubbled with 95% O$_2$/5% CO$_2$, and then transferred to room temperature in the same solution. Slices were visualized using a 40 × water immersion objective on an upright microscope (Scientifica) equipped with a custom-made contrast imaging gradient (Dodt optics), a mercury burner, and a Texas Red filter set. In most experiments, patch pipettes were filled with the following (mM): 150 K-gluconate, 1 MgCl$_2$, 1.1 EGTA, 5 HEPES and 10 phosphocreatine, pH 7.2 and 300 mOsm. In experiments to measure NMDAR EPSCs, we used a pipette internal with the following (mM): 97.5 Cs-gluconate, 17.5 CsCl, 8 NaCl, 10 BAPTA, 10 HEPES, 2 MgATP, 0.3 Na$_3$GTP, 7 phosphocreatine and 5 QX-314, pH 7.2 and 290 mOsm. Biocytin (0.2%) was included in the pipette in some experiments for morphological visualization after recording using streptavidin conjugated to Alexa Fluor 647. Synaptic responses were evoked using patch pipettes filled with extracellular solution (100 µs; 2–12 µA or 100–300 µA). All recordings were done at room temperature and at a holding potential of −70 mV unless otherwise noted. EPSC latencies were measured from the onset of the stimulus artifact to the onset of the EPSC. Miniature EPSCs were recorded in 0.5 µM TTX. Series resistance was uncompensated (10–25 MΩ) and experiments were discarded if substantial changes (>20%) were observed. Voltages were not corrected for junction potentials and currents were filtered at 2 kHz and sampled at 10 kHz (MultiClamp 700A; Molecular Devices). Action potential threshold was detected when the slope exceeded 10 mV ms$^{-1}$ and the peak was measured from the threshold. Number of spikes was calculated from the train of action potentials elicited by the highest current step (90–130 pA). Input resistance ($R_{input}$) was obtained from hyperpolarizing current injections of 20 pA for mature GCs and 10 pA for immature GCs. Bridge balance was automatically adjusted in the Multiclamp commander. CA recordings were performed with a patch pipette filled with artificial cerebral spinal fluid (ASCF) in voltage-clamp mode at current = 0 pA. Recordings were acquired with pClamp10 (Molecular Devices) and analysed using Axograph X (Axograph Scientific). Drugs and chemicals were obtained from Sigma-Aldrich, Tocris Bioscience, or Ascent Scientific.

Confocal images were taken from biocytin-filled mature and newborn GCs in acute slices after overnight fixation. GC morphology was reconstructed from image stacks using the tracing program Neurolucida (MicroBrightfield).

**Statistical analysis.** Data were expressed as mean ± s.e.m. To minimize type I error, we set the α-level at 0.05 and accepted significant results with $P < 0.05$ for all statistical tests. Normality was estimated using Shapiro–Wilk test, Kolmogorov–Smirnov test and Llliefors test. When data sets satisfied normality criteria, we used two-tailed $t$-tests or two-way analysis of variance repeated-measures to evaluate differences among two or multiple samples, respectively (Statistica, StatSoft and GraphPad Prism). We evaluated the effect of drug (gabazine and control), the difference between GCs at the same or different ages (mature or immature) and between multiple pathways (MPP/LPP and MEC/LEC), across increasing stimulus intensities. The $F$ values indicate the significant difference concerning the main factor (drug, cell age, pathway and stimulus) and their interaction; *post hoc* analyses were made with Tukey's tests. The homogeneity of the variance between populations was verified by Levene's test. In some cases where normality could not be verified, we used nonparametric tests: Wilcoxon for paired, Kolmogorov–Smirnov for unpaired data. Fits of EPSC progressive block were made by two exponential or linear functions, and the goodness of the fit was estimated by calculating the $\chi^2$ (OriginPro). The best fit was obtained by minimizing the mean square error between the data and the curve (Levenberg–Marquardt algorithm).

**Model methods.** *Estimating perforant path connection densities from slice experiments.* Dentate GCs receive thousands of excitatory synaptic inputs from the lateral and medial EC, however, the number of viable synapses that are potentially activated using focal stimulation in the slice preparation is a small fraction of the total number. To estimate the number of active synaptic inputs onto mature and immature GCs in slices and the degree of overlapping synaptic inputs, we fit a basic statistical model to the data shown in Fig. 6. Given a pair of neurons, the number of expected 'shared' inputs ($N_{shared}$; that is, source fibres both neurons receive an

input from) and 'independent' inputs ($N_{ind}$; that is, source fibres unique to one of the neurons) can be given by

$$N_{shared} = \frac{N_{total}}{N_\infty} \times N_{total} \quad (1)$$

$$N_{ind} = N_{total} - N_{shared} \quad (2)$$

where $N_{total}$ is the total number of functional synapses on a GC (from that projection) and $N_\infty$ is the total number of potential input fibres. Equations (1) and (2) simply mean that if two neurons are each sampling a fraction of potential input fibres, then the number of shared inputs is the same as the overall sampling density. For example, if $N_\infty$ is 500 and $N_{total}$ is 50, then they would be expected to share 10%, or 5, of their input fibres (with the 45 inputs on each neuron not overlapping). In contrast, if the neurons sample a much higher density ($N_{total}$ is 200), then the inputs would overlap by 40%.

$N_{total}$ and $N_\infty$ (and by extension $N_{shared}$ and $N_{ind}$), cannot be measured directly, so they must be numerically fit to the measurements in Fig. 6, which indicate the correlation of active synaptic inputs in pairs of simultaneously recorded neurons in response to increasing numbers of active perforant path fibres. Specifically, if we ignore magnitude of response and simply ask whether perforant path stimulation evokes an EPSC in both neurons (the output of at least one active fibre), the probability that both neurons respond is a function of both their independent and shared input fibres. If we assume that stimulation of the fibre bundle activates a fraction $p$ of the total inputs, we can derive the probability that both GCs receive an active fibre by the following equation based on binomial probabilities (which compute the probability that a random stimulation will activate fibres innervating both neurons by chance):

$$P_{not\ shared} = (1-p)^{N_{shared}} \quad (3)$$

Where $P_{not\ shared}$ is the binomial probability that the activation of a proportion of $p$ inputs, given $N_{shared}$ chances, would fail to evoke EPSCs in both neurons. Similarly,

$$P_{not\ ind} = (1-p)^{N_{ind}} \quad (4)$$

is the probability ($P_{not\ ind}$) that the proportion of $p$ inputs, given $N_{ind}$ chances, fails to evoke an EPSC in one of the neurons. Ultimately, we do not care if one neuron has an EPSC if the other does not; rather we care about whether both get EPSCs, thus

$$P_{both\ ind} = (1 - P_{not\ ind})^2 \quad (5)$$

where $P_{both\ ind}$ is the chance that two neurons receive EPSCs from at least one independent fibre by chance. Following, the probability that both neurons are not activated by independent fibres, $P_{not\ both\ ind}$, can be given by

$$P_{not\ both\ ind} = 1 - P_{both\ ind} \quad (6)$$

Finally, we are concerned with the probability that both neurons receive EPSCs simultaneously, since that is what we can measure. To compute this, we must subtract from one the mutual probability that the two neurons are neither activated by independent inputs nor by shared inputs:

$$P_{overlap} = 1 - P_{not\ both\ ind} \times P_{not\ shared} \quad (7)$$

Substituting equations (3)–(6) into equation (7) gives the following expanded form

$$P_{overlap} = 1 - (1 - (1 - (1-p)^{N_{ind}})^2) \times (1-p)^{N_{shared}} \quad (8)$$

In this equation, $P_{overlap}$ is measurable for different experimental multiples of $p$. Notably, we do not know the absolute value of $p$ for any given stimulation, but we can assume that if we are far enough below saturation, increases in the experimental stimulation intensity yield a proportional increase in proportion of input fibres activated. As a result, the proportion of synaptic inputs activated for the minimal experimental stimulation in Fig. 6 is considered to be the $p$ parameter (with higher amplitude stimulations resulting in a multiple of $p$), with $N_{ind}$ and $N_{shared}$ being the other parameters necessary to fit.

On the basis of Fig. 7a, we used the constraint that the ratio of intact synapses on young neurons to mature neurons ($N_{total-young}/N_{total-mature}$) is 0.35. Further, we constrained the $N_{ind,mature}$ to be no more than five times $N_{shared,mature}$. Our goal was to minimize the squared error between equation (8)'s estimates for overlapping outputs between two neurons and our experimental measurements in Fig. 6b, per the following equation

$$err = \sqrt{\sum_{stim} (P_{estimated,stim} - P_{measured,stim})^2} \quad (9)$$

where $P_{estimated,stim}$ is the output of equation (8) and $P_{measured,stim}$ refers to the measured overlap for a given stimulation level in Fig. 6b.

Since the binomial relationship in equation (8) was not well suited for an analytical optimization of the parameters that globally minimize the error in equation (9), we used a Monte Carlo exploration of the space to find combinations of $N_{ind}$, $N_{shared}$, and $p$ that gave good fits. We structured our Monte Carlo search to have 250 000 combinations of the three independent parameters: $0.001 < P < 0.005$, $20 < N_{ind,\ mature} < 100$ and $100 < N_{shared,\ mature} < 500$, identifying which set of parameters produced a good fit per equation (9). Notably, there were a number of solutions with approximately equivalent errors for which we selected $P = 0.0039$;

$N_{\text{ind,mature}} = 182$; $N_{\text{shared,mature}} = 37$, with a cumulative error (when compared with both immature and mature physiology data) of 0.18. Importantly, our results and interpretation are robust to these different minima; we tested several other effective fits to equation (9), even for cases outside our above search constraints, and reliably observed comparable results to those we selected here. Results from progressive samples of the top 20% of parameter combinations are shown in Supplementary Table 1, with the selected fit shown in bold (for each row, the percentile shows where the parameter set ranked among the fits).

**Simple neural network model.** We generated a simple perceptron-based neural network model of the EC to DG circuit[21]. This model clearly is a considerable abstraction from the biological system and lacks spiking and long-timescale dynamics; however, it can illustrate how neuron variation can influence output correlations of the system. The model consisted of 13,000 GC neurons and 1,300 EC neurons. The 1,300 EC neurons scale was selected to be comparable to the estimate of preserved inputs available within a slice, and the 13,000 GCs scale was selected to allow us to investigate the large, close to 1:10, expansion ratio from EC to DG. Each GC neuron was either considered mature or immature and randomly connected to neurons in the source EC population based on the frequencies determined above. Any connection resulted in a synapse of weight 1, and there was no learning or inhibition in the network. Neurons are considered on (activity = 1) if their inputs are above their threshold, otherwise they are off (activity = 0).

For each trial, a fraction of EC neurons, $EC_{\text{act}}$, was randomly activated ($EC_{\text{ECact}} = 1$), while all other neurons were off ($EC_{\sim \text{ECact}} = 0$). The downstream GC neurons were then considered active if their input surpassed their threshold, which was defined as 20% of their synaptic inputs being co-active at any given time step. This rule allows immature neurons to be active with fewer active inputs, in accordance with their higher intrinsic excitability.

$$GC_{\text{input}} = EC \times W_{\text{ECtoGC}} \qquad (10)$$

$$GC_{\text{output}} = \begin{cases} 1 & \text{if } GC_{\text{input}} \geq 0.2 \times N_{\text{synapses}} \\ 0 & \text{if } GC_{\text{input}} < 0.2 \times N_{\text{synapses}} \end{cases} \qquad (11)$$

We tested each network on 100 sets of random EC inputs, and then computed the average overlap between GC outputs, which is given by

$$DG_{\text{NDP}} = \frac{1}{50 \times 99} \times \sum_{i=2}^{100} \sum_{j=1}^{i-1} \frac{GC_{\text{output},i} \cdot GC_{\text{output},j}}{\|GC_{\text{output},i}\| \|GC_{\text{output},j}\|} \qquad (12)$$

To assess how levels of neurogenesis affect the dynamic range of permissible EC inputs, we ran 101 neurogenesis levels (networks containing from 0–100% of immature neurons, in 1% increments) with 481 EC levels (0.10–0.22 of EC inputs in 0.0025 increments). Dynamic range for each simulation was measured by subtracting the EC level that provided at least a NDP of 0.005 from the EC level that provided at least an NDP of 0.05. Five simulations were run for each NG level and the standard deviations of the dynamic ranges over the five runs for each NG value were always less than 5% of the mean dynamic range.

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

## Acknowledgements

We thank members of the Wadiche Labs for helpful discussion throughout this project, and Antoine Madar and Dr Mathew Jones for comments on the manuscript. This work was supported by NIH NS064025 (L.O.-W.), NIH NS065920, NSF 1539034 (J.I.W.) and NIH NS047466. C.T.K. is supported by NIH MH105416 and NS078192. J.B.A. is supported by Sandia National Laboratories' Laboratory Directed Research and Development (LDRD) program. Sandia National Laboratories is a multi-program laboratory managed and operated by Sandia Corporation, a wholly owned subsidiary of Lockheed Martin Corporation, for the US Department of Energy's National Nuclear Security Administration under contract DE-AC04-94AL85000.

## Author contributions

C.V.D., R.P., J.B.A., J.I.W. and L.O.-W. designed, performed or analysed the experiments and simulations. C.K. provided technical expertise and reagents. C.V.D., J.I.W. and L.O.-W. wrote the manuscript. All authors contributed scientific insights and provided critical readings of the manuscript.

## Additional information

**Competing financial interests:** The authors declare no competing financial interests.

