## [Peer Review file · Nature Communications]

Reviewers' Comments:

Reviewer #1 (Remarks to the Author)

The revised version of the manuscript has been substantially improved and it seems now suitable for publication.

Reviewer #3 (Remarks to the Author)

There are positive changes in the current version compared to previous versions. Now the authors put less emphasis on the elusive pattern separation function at cellular level. The message of the manuscript is much clearer as reflected, for example, by the more specific title or by the more comprehensive introduction. The manuscript has very clear and well-explained aims and these are completely answered by the results. The experimental approach and data are of high quality and the authors fully answered our previous concerns and questions by providing an additional "reviewer-only" figure about the variable noise of the recordings, by adding more original traces to the figures (e.g. Fig.3) and by showing the baseline period in the MK-801 experiments.

The results section is also improved by the additional explanations about the observations and data.

In summary, we believe that this manuscript should be accepted as is. (the very minor points can be easily addressed by the authors).

Minor issues:

In Figure 1c the spikes in the mature GC are much larger compared to immature cell. The threshold of spikes also appears to be different (although the voltage may not be aligned in the two sets of traces). Such large differences in the spike parameters are not consistent with previous publications and with the data on the graphs in the right panels. Probably those traces which were included in the previous version of the manuscript are more representative for the results.

Typo: p.5. "small-evoked EPSCs"

Response to Reviewers

Reviewer #3 (Remarks to the Author)

There are positive changes in the current version compared to previous versions. Now the authors put less emphasis on the elusive pattern separation function at cellular level. The message of the manuscript is much clearer as reflected, for example, by the more specific title or by the more comprehensive introduction.

The manuscript has very clear and well-explained aims and these are completely answered by the results. The experimental approach and data are of high quality and the authors fully answered our previous concerns and questions by providing an additional "reviewer-only" figure about the variable noise of the recordings, by adding more original traces to the figures (e.g. Fig.3) and by showing the baseline period in the MK-801 experiments.

The results section is also improved by the additional explanations about the observations and data.

In summary, we believe that this manuscript should be accepted as is. (the very minor points can be easily addressed by the authors).

Minor issues:

In Figure 1c the spikes in the mature GC are much larger compared to immature cell. The threshold of spikes also appears to be different (although the voltage may not be aligned in the two sets of traces). Such large differences in the spike parameters are not consistent with previous publications and with the data on the graphs in the right panels. Probably those traces which were included in the previous version of the manuscript are more representative for the results.

We have switched the traces to more representative examples.

Typo: p.5. "small-evoked EPSCs"

Corrected.